# Probabilistic analysis of COVID-19 patients' individual length of stay in Swiss intensive care units

Alexander Henzi [1], Gian-Reto Kleger[2], Matthias P. Hilty [3,4], Pedro D. Wendel Garcia[3,4], Johanna F. Ziegel [1] *, on behalf of RISC-19-ICU Investigators for Switzerland[¶]

**1** Institute of Mathematical Statistics and Actuarial Science, University of Bern, Bern, Switzerland, **2** Division of Intensive Care Medicine, Cantonal Hospital, St.Gallen, Switzerland, **3** The RISC-19-ICU Registry Board, University of Zurich, Zürich, Switzerland, **4** Institute of Intensive Care Medicine, University Hospital of Zürich, Zürich, Switzerland

[¶] Membership of the RISC-19-ICU Investigators for Switzerland is provided in the Acknowledgments.
* johanna.ziegel@stat.unibe.ch

**Data Availability Statement:** Any intensive care unit or other centre treating critically ill COVID-19 patients is invited to join the RISC-19-ICU registry

## Abstract

### Rationale

The COVID-19 pandemic induces considerable strain on intensive care unit resources.

### Objectives

We aim to provide early predictions of individual patients' intensive care unit length of stay, which might improve resource allocation and patient care during the on-going pandemic.

### Methods

We developed a new semiparametric distributional index model depending on covariates which are available within 24h after intensive care unit admission. The model was trained on a large cohort of acute respiratory distress syndrome patients out of the Minimal Dataset of the Swiss Society of Intensive Care Medicine. Then, we predict individual length of stay of patients in the RISC-19-ICU registry.

### Measurements

The RISC-19-ICU Investigators for Switzerland collected data of 557 critically ill patients with COVID-19.

### Main results

The model gives probabilistically and marginally calibrated predictions which are more informative than the empirical length of stay distribution of the training data. However, marginal calibration was worse after approximately 20 days in the whole cohort and in different subgroups. Long staying COVID-19 patients have shorter length of stay than regular acute respiratory distress syndrome patients. We found differences in LoS with respect to age categories and gender but not in regions of Switzerland with different stress of intensive care unit resources.

at https://www.risc-19-icu.net. While the registry protocol prevents the deposition of the full registry dataset in a third-party repository, analyses on the full dataset may be requested by any collaborating centre after approval of the study protocol by the registry board. Reproducibility of the results in the present study is ensured by providing underlying code and an adapted dataset to exemplarily test the later. The registry protocol and data dictionary is publicly accessible at https://www.risc-19-icu.net.

**Funding:** The author(s) received no specific funding for this work.

**Competing interests:** The authors have declared that no competing interests exist.

## Conclusion

A new probabilistic model permits calibrated and informative probabilistic prediction of LoS of individual patients with COVID-19. Long staying patients could be discovered early. The model may be the basis to simulate stochastic models for bed occupation in intensive care units under different casemix scenarios.

## 1 Introduction

During the COVID-19 pandemic, governments worldwide imposed severe restrictions on public life in order to limit the spread of the SARS-CoV-2 virus. A critical point in the decision making process was the limitation of beds in intensive care units (ICU) in order to adequately treat all severe cases of COVID-19. Many countries increased the number of ICU beds substantially at the onset of the crisis. A critical issue with severe COVID-19 disease is the frequent need for prolonged ICU treatment. For informed decision making it is important to quantitatively assess how long the patients are expected to be in an ICU.

At the example of Switzerland, we propose a prediction method for the individual length of stay (LoS) of patients in ICUs, and apply it to COVID-19 patients. The predictions are given for each patient based on covariates available within 24 hours after ICU admission. The method generates probabilistic predictions, that is, for each patient that enters the ICU, we provide a predictive cumulative distribution function (CDF) that comprehensively quantifies the uncertainty of the LoS at the time point of prediction. In particular, the predictive CDF allows to give prediction intervals with nay desired coverage probability. More precisely, the predictive CDF is an estimate of the conditional distribution of the LoS of the patient given covariates, which include age, gender, Simplified Acute Physiology Score (SAPS II) [1], and Nine Equivalents of nursing Manpower use Score (NEMS) (first shift) [2]. Fig 1 shows some predictive CDFs for randomly selected COVID-19 patients black, and true LoS as vertical lines. For each possible value $t$ of the LoS, the value of the predictive CDF, $F(t)$, gives the probability that the patient stays at most $t$ days in the ICU. Conversely, $1 - F(t)$ gives the probability that the patient stays longer than $t$ days in the ICU. For example, patient 1 had an LoS of 20 days. The predicted probability that the patient stays at most 20 days was 0.91, and the probability for a stay of at least 10 days was 0.26 (or 0.74 for at most 10 days). Patient 4 stays longest in the ICU. This is in agreement with the predictive CDFs, since for all possible $t$, the probability of staying longer than $t$ is highest for patient 4. The waves in the curves are explained by the fact that patients have a higher possibility to leave the ICU at certain times of the day, and a lower at others.

Probabilistic predictions allow to assess the uncertainty of the LoS comprehensively. Therefore they are preferable to forecasts for the mean or median LoS only. Their usefulness is illustrated by the following examples. The probabilistic LoS predictions allow to derive probabilistic forecasts for the number of patients who are still at the ICU at a certain day in future. This may be useful for planning purposes. For a single patient admitted today with predictive LoS distribution $F$, the probability that the patient is still at the ICU after $t$ days equals $1 - F(t)$. From the probabilities for single patients, one may compute (with statistical software) the probability that any given number of patients is still at the ICU after $t$ days. This allows to answer questions like 'How likely is it that there are at least two free beds in five days?' or 'What is the smallest number of patients we expect to stay until next week with a high probability (say, 90%)?'. The LoS forecasts, and so also the answers to such questions, take into account the individual characteristics of the patients currently at the ICU. The probabilistic LoS

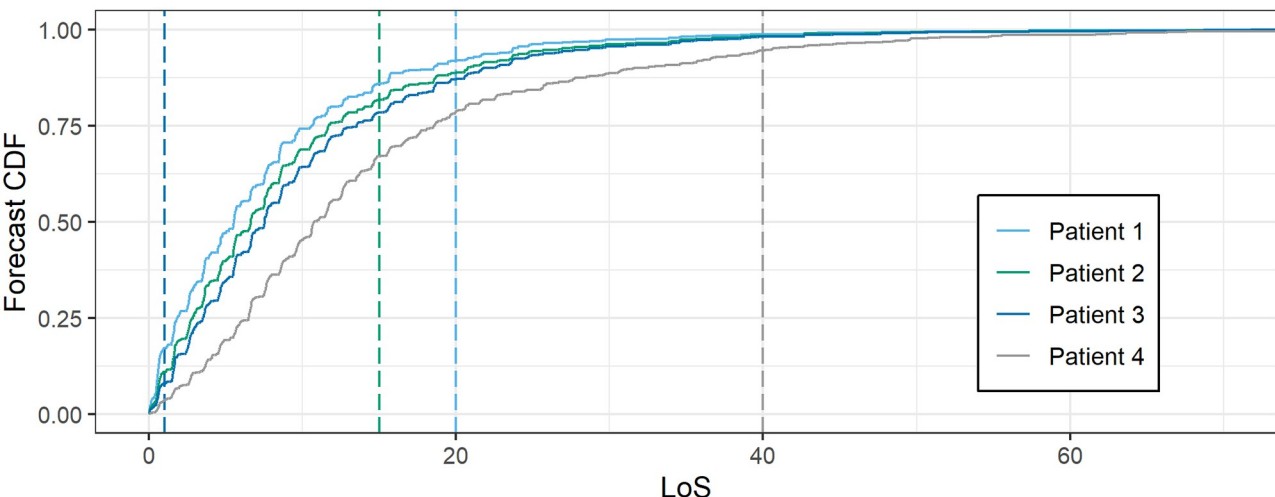

**Fig 1. Predictive CDFs for the LoS of some COVID-19 patients with corresponding realizations as a vertical line.** Four patients were drawn at random. The four wavy lines represent their predictive CDFs for the LoS based on covariates that are available at most 24 hours after ICU admission, that is, for each value $t$ of the LoS on the horizontal axis, the curve gives the probability that the respective patient stays at most $t$ days in the ICU. The vertical dashed lines represent the actually observed values of the LoS of the patients, which are unknown at the time of prediction. The larger the increase of the CDF on a given interval on the horizontal axis, the higher the probability of observing an LoS in this interval. For example, the predicted probability for the LoS of patient 1 being between 0 and 5 days is 0.47, whereas this probability is 0.40 for patient 2, 0.35 for patient 3, and only 0.19 for patient 4. The CDF of patient 4 lies substantially below the CDFs of the other patients which is in agreement with patient 4 having the longest realized LoS.

predictions also allow to give alerts for patients that are likely to stay unusually long in the LOS. For example, fix a threshold of $x$ days, say $x = 25$, and give an alert if the probability that the patient stays longer than $x$ days exceeds, say, 90%. That is, if $1 − F(x) > 90\%$, where $F$ is the predictive LoS distribution of a specific patient.

For planning of normal ward and intermediate care unit to ICU patient flows, such information is key to allow optimized resource allocation. On a larger scale, one could plan regional patient allocations to multiple hospitals based on such algorithms. The current health care crisis has emphasized the importance of patient flow logistics, and informative predictions of LoS are essential for this purpose.

It is documented in the literature that the prediction of the LoS at the patient level is difficult, and none of the available prediction models is providing satisfactory forecasts [3] with a possible exception being the complex models presented in [4, 5] for the purpose of benchmarking. Furthermore, the focus has almost exclusively been on only point predictions for the mean LoS, which is not ideal given that the LoS distribution is heavily skewed.

Recently, methodological progress has been made by Ziegel's group [6]: Based on data in the format of the Minimal Dataset of the Swiss Society of Intensive Care Medicine (MDSi), it is possible to give skillful and calibrated probabilistic predictions for the LoS of patients in ICUs 24h after their admission. In particular, the predictions for the probability of exceedance of the LOS over a certain threshold is shown to be reliable. The proposed method is semi-parametric, which makes it highly adaptive to the shape of the conditional LoS distributions. However, it requires large training data sets. The currently available data on COVID-19 patients in Swiss ICUs is (fortunately) not sufficient. Therefore, we suggest to borrow strength from the MDSi in order to predict the conditional LoS of COVID-19 patients.

The LoS of a patient in an ICU does not only depend on their physical condition but also on the characteristics and policies of the ICU. Even within a small country such as Switzerland such differences can be observed [6]. We restrict the analysis in this paper to Switzerland but

the methodology can be adapted to other countries given sufficient data is available. We use the prediction method for the LoS to analyze the characteristics of the LoS of COVID-19 patients with respect to age differences, and gender differences. Since some parts of Switzerland were hit harder by the pandemic than others, we also use the predictions to analyse regional differences in the LoS.

## 2 Patients and methods

### 2.1 RISC-19-ICU and MDSi

Risk Stratification in COVID-19 patients in the Intensive Care Unit (RISC-19-ICU) registry, is a collaborative effort with the participation of a majority of the Swiss ICUs to provide a basis for decision support during the ongoing public health crisis, endorsed by the Swiss Society of Intensive Care Medicine (https://www.risc-19-icu.net/) [7, 8]. ICU data were reported on a daily basis, including near real-time data on LoS. The registry was deemed exempt from the need for additional ethics approval and patient informed consent by the ethics committee of the University of Zurich (KEK 2020-00322, ClinicalTrials.gov Identifier: NCT04357275). Fully anonymized datasets, in regard to Swiss law, were collected using a secure REDCap infrastructure provided by the Swiss Society of Intensive Care Medicine.

557 critically ill patients with COVID-19 that have been admitted to an ICU in Switzerland have entered the registry as of the snapshot date, June15, 2020, 481 of which have already been dismissed from the ICU or have died, that is for 86.36% of the patients the LoS is available. There are 18 patients for which one or more of the covariates are not available. Overall, covariates and LoS observations are available for 473 patients, and we call these the COVID-19 dataset. Censoring is a non-trivial problem in the COVID-19 dataset and we address this issue in detail in Section A of S1 Appendix.

The Minimal Dataset of the Swiss Society of Intensive Care Medicine (MDSi) has been introduced in 2005 and contains fully anonymized key data of the entire number of ICU patients in certified Swiss ICU's (https://www.sgi-ssmi.ch/de/datensatz.html). In addition to demographic data, the MDSi includes SAPS II as initial illness severity score and NEMS per nursing shift as a workload score.

Because almost any patient with severe COVID-19 disease presents chiefly like acute respiratory distress syndrome (ARDS), the training data consists of all patients in the MDSi with the diagnosis of ARDS which were admitted to Swiss ICUs in the years 2012 to 2018. Of the 2411 admissions, 856 were excluded because they satisfy one or more of the following criteria: missing or implausible values for SAPS II or NEMS (135), age younger than 16 (5), admitted with burns as initial diagnosis (3) or undergoing transplant operations 24 hours before or after ICU admission (8), readmissions (132), and patients admitted from ICUs or transferred to other ICUs (580). The exclusion of patients transferred from or to ICUs is because their LoS is incomplete and therefore not suitable for prediction. For the LoS predictions, admissions are standardized to a common admission time at midnight, in order to recover patterns in the ICU discharge times [6]. As a consequence, 99 patients had to be excluded because they did not stay in the ICU at least until midnight of the admission date. After exclusions, the training dataset consists of 1555 observations.

Concerning the covariates that are available for prediction, the possibilities are limited to covariates that are available in the COVID-19 dataset and the training data in the same format. Clear choices are the gender and age of patients. Furthermore, we have included SAPS II and the NEMS of the first ICU shift as covariates since they are informative for the LoS [9–11].

## 2.2 Statistical methods

Distributional Index Models (DIMs) have been introduced in [6]. They are semi-parametric models for distributional regression building on isotonic distributional regression (IDR) introduced in [12, 13]. A distributional regression model allows to estimate the full conditional distribution of the LoS given covariates. For the DIM used in this paper, we use a parametric model for a real-valued index function $\alpha$, the DIM index, that depends on gender $g$, age $a$, SAPS II $s$, and NEMS $m$, that is

$$\alpha(g, a, s, m) = \beta_0 + \beta_1 \mathbf{1}\{g = \text{male}\} + \text{cr}_1(a) + \text{cr}_2(s) + \text{cr}_3(m),$$

where $\beta_0$ is the intercept, $\beta_1$ the coefficient for gender, and $\text{cr}_1$, $\text{cr}_2$, $\text{cr}_3$ are penalized cubic regression splines for the continuous variables age, SAPS II and NEMS; see the documentation of the mgcv package for details about the penalization. The model is fitted on the transformed LoS log(LoS+ 1). The log-transformation decreases the skewness of the data, while the addition of the constant 1 makes the resulting distribution more symmetric [6].

Furthermore, we assume that for the probability of the LoS $Y$ of a randomly selected patient with covariates $(G, A, S, M) = (g, a, s, m)$ it holds that

$$\mathbb{P}(Y \leq y | (G, A, S, M) = (g, a, s, m)) = F_{\alpha(g,a,s,m)}(y), \quad \text{for all } y \in \mathbb{R} \quad (1)$$

with a family $(F_v)_{v \in \mathbb{R}}$ of stochastically ordered CDFs, that is $F_v(y) \leq F_w(y)$ for all $y \in \mathbb{R}$ if $v \geq w$.

We randomly split the training data in two and estimate $\alpha$ by $\hat{\alpha}$ on the first half. Given $\hat{\alpha}$, we use the second half of the training data to estimate $F_v$ using IDR. In order to make the estimation procedure less dependent on the splitting of the training data, we use repeat this procedure 100 times and average the resulting estimated distributions to obtain our final estimate $\hat{F}\hat{\alpha}$.

There are dependencies between the covariates age, SAPS II and NEMS but we argue that it is still useful to include all of them in the model. The variable age is contained in SAPS II as a discretized effect with 6 levels. Age enters the model as a cubic regression spline with sufficiently high dimension, manually removing the age variable from SAPS II would essentially correspond to a basis transformation of the model and not affect the prediction results. The information provided by the NEMS is not redundant to SAPS II. NEMS is a crucial variable for COVID-19 patients since it contains information on the ventilation status, therapy with cardiovascular drugs and renal replacement treatment, which are not in the SAPS II. More precisely than the SAPS II, the NEMS reflects the actual therapeutic intensity a patients needs, and it is therefore likely to be one of the earliest markers for LoS.

Probabilistic predictions should be calibrated and sharp [14]. We assess probabilistic calibration by Probability Integral Transform (PIT) histograms, and use Pearson's chi-square test with 10 bins to test for uniformity. Marginal calibration is checked by comparing average predicted CDFs to empirical CDFs (ECDFs). Sharpness is assessed using the Continuous Ranked Probability Score (CRPS) and predictive power is compared with a Diebold-Mariano test based on the CRPS, see Section B of S1 Appendix.

The implementation is done in R 4.0 [15] using the packages mgcv [16] for the estimation of the index function, and isodistrreg for isotonic distributional regression [12]. Sample data and code are provided in the supplement S1 Code of this article.

# 3 Results

## 3.1 General

Summary statistics for the COVID-19 dataset and the training data are given in Table 1. The figures are correct for the June 15, 2020, snapshot. The proportion of men in the COVID-19 dataset is higher than in the training data set. The age structure of both datasets is similar with COVID-19 patients being slightly younger on average. COVID-19 patients generally have a higher NEMS in the first shift. The median and mean SAPS II is similar in both datasets.

Fig 2 provides a quantitative comparison of the LoS in the COVID-19 dataset and the training data. Panel (a) shows that the probability $\mathbb{P}(Y \geq y)$ of the LoS exceeding a fixed threshold $y$ is larger for COVID-19 patients than in the training data up to about $y = 30$ days, and afterwards the relationship is reversed.

This observation does not exclude the possibility that given the covariates $(G, A, S, M)$ for an individual patient, the conditional distribution of the LOS can be predicted well using the training data. Panel(b) of Fig 2 shows that the individual predictions are reasonable and are marginally calibrated up to about 25 days. The tail of the average forecast distribution is heavier than the tail of the empirical distribution of the COVID-19 dataset, meaning that very long LoS are less likely in the COVID-19 dataset.

The DIM predictions for the LoS of the COVID-19 patients have an average CRPS of 5.29 compared to 5.69, which is the average CRPS when predicting the LoS of the COVID-19 patients with the ECDF of the training data, that is, for all patients, independently of the covariates, the LoS is predicted by using the distribution of all the LoS values in the training data. This difference is highly significant with a p-value of less than $5 \cdot 10^{-4}$. This shows that the DIM predictions are significantly more informative than the ECDF forecast. The DIM predictions show better calibration than the ECDF predictions, see S3 Fig in S1 Appendix. Uniformity of the PIT is rejected for the ECDF forecasts (p-value $< 10^{-4}$). For the DIM forecasts, uniformity of the PIT is not rejected (p-value: 0.384).

## 3.2 Age differences

Fig 3(a) gives the empirical CDFs of COVID-19 patients grouped by age. Young patients, less than 40 years, and very old patients, greater than 80 years have much shorter LoS than patients between 40 and 80. Patients between 40 and 65 tend to have a shorter LoS than patients between 65 and 80 except in cases of long LoS beyond 30 days. In Fig 3(c) the empirical CDFs

**Table 1. Summary statistics of COVID-19 dataset and training data.**

| Variable | Data | Q25 | Median | Mean | Q75 | P-value |
|----------|------|-----|--------|------|-----|---------|
| Age | training | 55.0 | 67.0 | 63.8 | 75.0 | $4.04 \cdot 10^{-3}$ |
| | COVID-19 | 55.0 | 63.0 | 63.0 | 72.0 | |
| LoS | training | 4.5 | 9.1 | 12.4 | 15.8 | $5.79 \cdot 10^{-5}$ |
| | COVID-19 | 5.0 | 12.0 | 13.9 | 19.0 | |
| NEMS | training | 18.0 | 27.0 | 28.6 | 34.0 | $<1.0 \cdot 10^{-16}$ |
| | COVID-19 | 32.0 | 32.0 | 33.2 | 39.0 | |
| SAPS II | training | 35.0 | 46.0 | 48.5 | 59.0 | $1.39 \cdot 10^{-1}$ |
| | COVID-19 | 29.0 | 50.0 | 44.9 | 58.0 | |
| Gender | training | Male: 61.9% | | Female: 38.1% | | $1.66 \cdot 10^{-8}$ |
| | COVID-19 | Male: 75.9% | | Female: 24.1% | | |

P-values are for two-sided Wilcoxon's rank sum test for continuous variables and Fisher's exact test for gender.

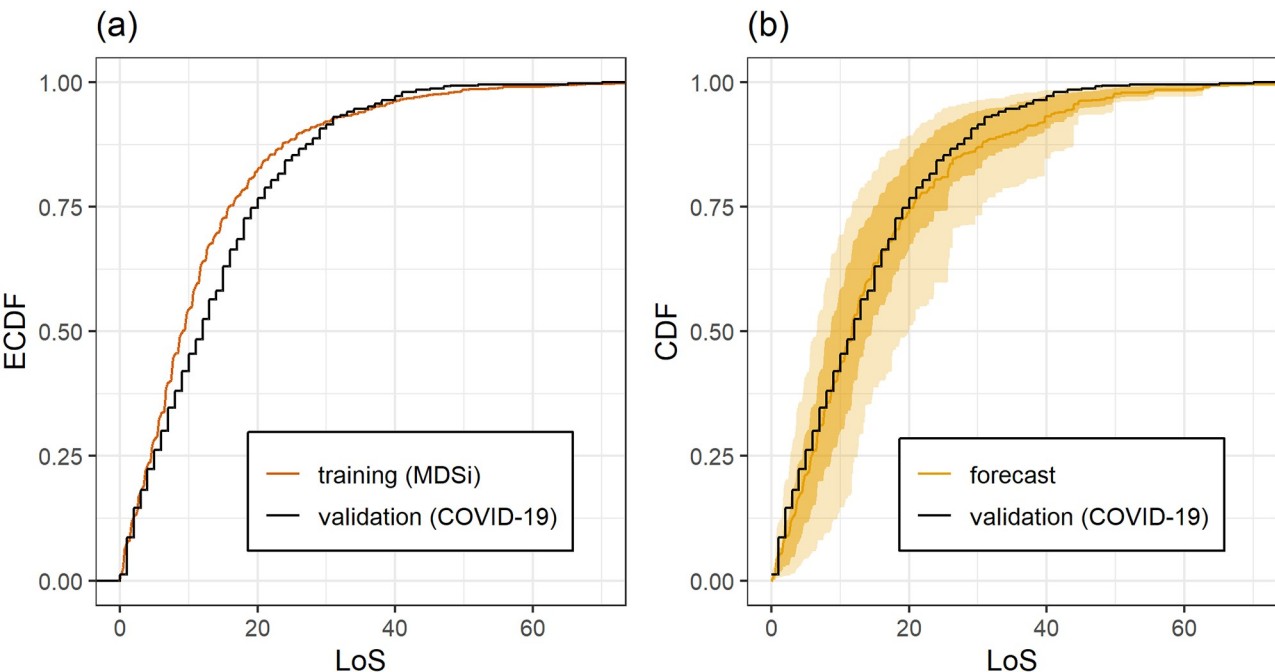

**Fig 2.** (a) EmpiricalCDF of the LoS in training and validation dataset. (b) Empirical CDF of the LoS in the validation dataset (black step function black, same as in panel (a)) and average LoS forecast for the COVID-19 patients (orange curve). Shaded areas show the pointwise 25% and 75% (10% and 90% for the outer bounds) quantiles of the predictive CDFs. For the average LoS forecast, the predictive CDFs of the COVID-19 patients are averaged pointwise, that is, the curves show the vertical average of the predictive CDFs for all patients in the COVID-19 dataset. The computation of the aggregated LoS CDFs is demonstrated in the sample code in S1 Code. The predictions take individual patient covariates into account and this allows to mitigate some differences between training and validation data observed in panel (a); for further discussion see text.

are compared to the predictions based on the training data. The predictions for patients younger than 40 seem reasonable but their quality is hard to judge given the small sample size of this group in the COVID-19 dataset. For patients older than 80, the predicted LoS is longer than observed, but again, a definite statement should not be made due to small sample size. For patients between 40 and 65, marginal calibration is good until about 18 days. For higher thresholds, a longer LoS is predicted than observed. For patients between 65 and 80 years, the predictions give too much weight to LoS shorter than 25 days, and substantially overestimate the LoS beyond 25 days. Fig 3(b) shows that the training data leads to predictions of shorter LoS for patients younger than 40 and older than 80. In contrast to the COVID-19 data, the predicted LoS for patients between 65 and 80 is shorter than for patients between 40 and 65.

### 3.3 Gender differences

Fig 4(a) shows the empirical CDF of COVID-19 patients grouped by gender. Female patients show a slightly shorter LoS. The deviations of the predicted LoS from the observed LoS for male and female patients is displayed in Fig 4(c). Qualitatively the differences are similar with a slightly worse agreement of predictions and observations for female patients. The average predictive distributions for male and female patients are displayed in Fig 4(b). The predictions show a clear difference depending on gender with the same order as the COVID-19 data in that the LoS for women tends to be shorter than the one for men. However, the difference in average predicted LoS CDF is larger than the difference in ECDF based on the COVID-19 data.

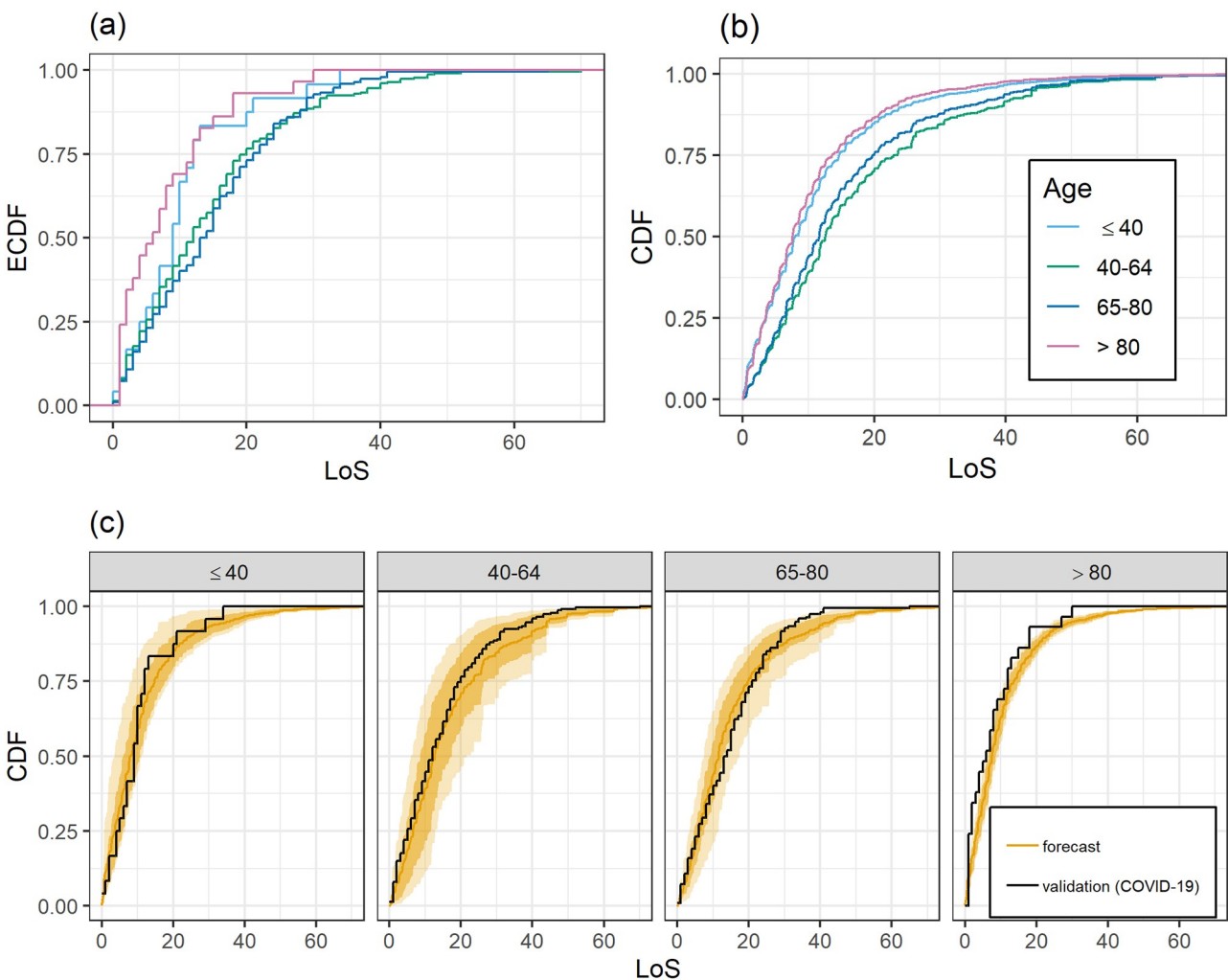

**Fig 3.** Depending on age: (a) Empirical LoS distributions of COVID-19 patients. (b) Average DIM forecasts for COVID-19 patients. (c) Empirical LoS distributions of COVID-19 patients and corresponding DIM forecasts. DIM forecasts are as in Fig 2.

In order to gain some insight on the reasons for this effect, we checked if there is a significant difference in the LoS distribution of men and women in the training data. This is not the case. Furthermore, a comparison of the distribution of the DIM index computed for the men and women in the COVID-19 dataset shows that, indeed, the index values for women tend to be smaller than those for men, which explains the differences between the CDFs in Fig 4(b). In summary, it appears that a female patient with COVID-19 is likely to stay longer in the ICU than a similar female patient in the training data, whereas this effect is less pronounced for men.

### 3.4 Regional differences

We have split the COVID-19 dataset according to the location of the ICU within Switzerland. Region NE consisting of Northern and Eastern Switzerland and Region WT consisting of Western Switzerland and Ticino. Region WT was hit earlier and more severely by the COVID-19 crisis than Region NE. While ICU capacity limits were never reached in Region NE, ICU occupation was possibly critical in Region WT.

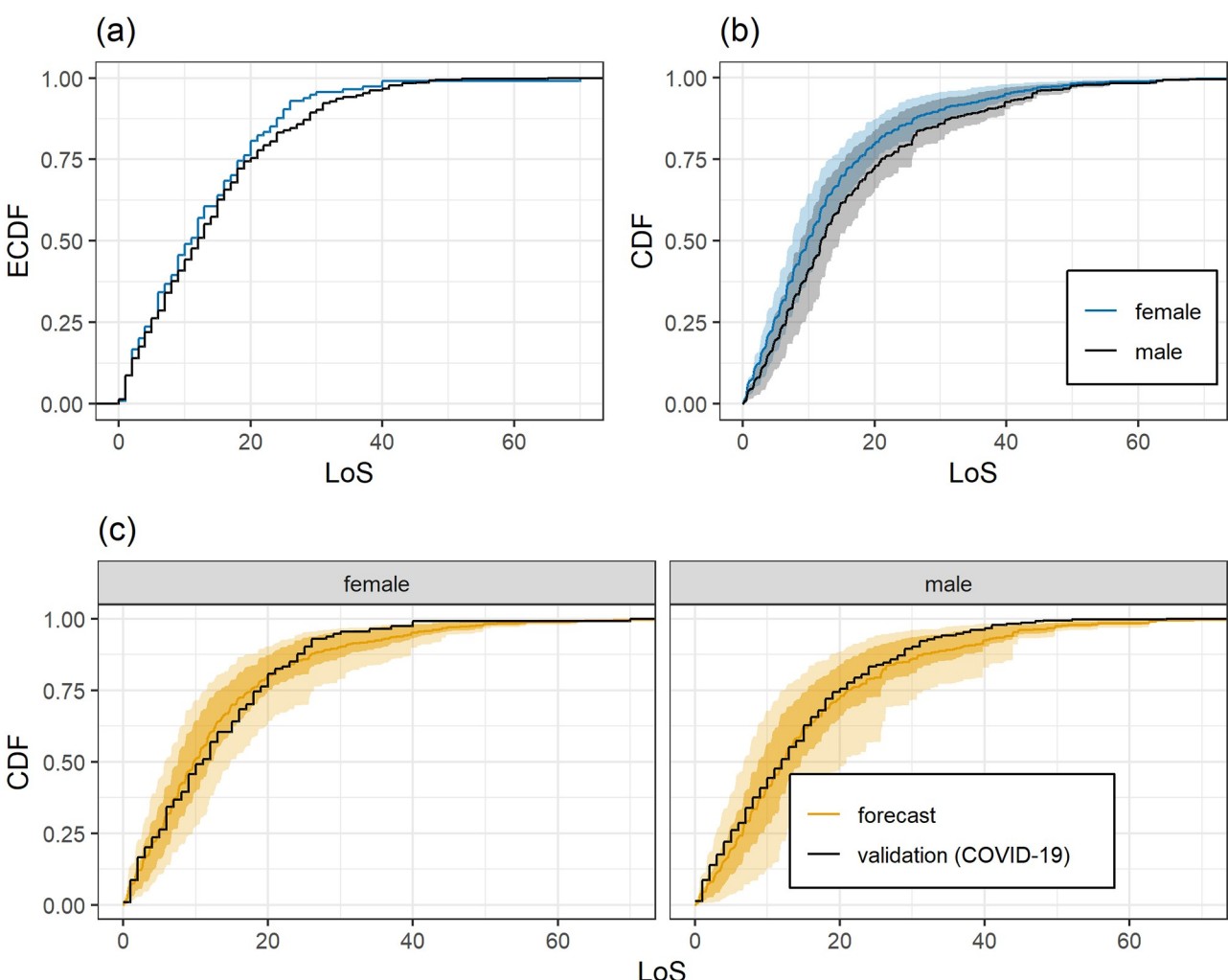

**Fig 4.** Depending on gender: (a) Empirical LoS distributions for COVID-19 patients. (b) Average DIM forecasts for COVID-19 patients. (c) Empirical LoS distributions of COVID-19 patients and corresponding DIM forecasts. DIM forecasts are as in Fig 2.

The LOS distribution of COVID-19 patients is similar in both regions. The null hypothesis of equal LoS distribution in both regions cannot be rejected (two-sample Kolmogorov-Smirnov test p-value: 0.510, Wilcoxon rank sum test p-value: 0.607), see also S4 Fig in S1 Appendix. Comparing the regional LoS distributions to the DIM forecasts for the regions, we obtain that both regions show the same pattern: Good marginal calibration until about 25 days and then shorter LoS of the COVID-19 patients in comparison to the DIM predictions, see S5 Fig in S1 Appendix. The differences in the predictions for both regions are small, see S6 Fig in S1 Appendix.

## 4 Discussion

We have applied a new semi-parametric model, a DIM, for probabilistic predictions for the LoS of COVID-19 patients in Swiss ICUs. The model is trained with data from the MDSi, namely with data of patients with ARDS. Validation of the model using the COVID-19 dataset shows that the predictions are probabilistically calibrated, marginally calibrated (except for the

tail of the distribution), and significantly more informative then an ECDF forecast based on the training data.

COVID-19 patients younger than 40 and older than 80 years tend to have a shorter stay in the ICU than the patient groups between 40–65 and 65–80 years. Predictions for patients older than 80 were longer than observed which could be an indicator of early treatment withdrawal in very old patients with severe COVID-19 disease. In the age groups 65-80 years, forecasts were shorter in the early phase than observations. This could be explained by prolonged recovery times compared with ARDS in elderly patients. The forecasts in both age groups (40–65 and 65–80 years) were longer after 25 to 30 days. In those patient groups, withdrawal of treatment is often executed after 20-30 days because of medical futility. The analysis of the LoS with respect to age suggests that there are differences between ARDS (training data) and COVID-19 in the sense that in terms of LoS COVID-19 patients might rather behave like slightly older ARDS patients keeping the other covariates fixed.

The difference between the LoS distribution of female and male COVID-19 patients is smaller than the difference between the predicted LoS distributions based on the training data, that is, non-COVID-19 patients with ARDS. For male patients the predictions agree better with the empirical distribution of observed LoS of the COVID-19 patients than for female patients. In terms of LoS, male COVID-19 patients behave more similarly to patients in the training data than female COVID-19 patients, making "longer than expected" LoS more likely for female than for male patients.

Despite the fact that the Western Switzerland and Ticino (Region WT) were hit earlier, and potentially less prepared for the COVID-19 crisis than Northern and Eastern Switzerland (Region NE), we do not see an impact on the LoS of COVID-19 patients.

There are somepossible shortcomings of our study. First, the training dataset is not on COVID-19 patients. Despite severe COVID-19 pneumonia behaving similar to ARDS, there are some important differences [17]. Furthermore, multiple organ involvement is frequent in severe COVID-19 disease [18, 19]. There have been discussions how and if classical ARDS and ARDS secondary to COVID-19 (C-ARDS) are different. Initially, substantial differences were postulated [20–22] but more recently consensus is growing that C-ARDS is most probably similar to classical ARDS in treatment intensity and therapeutic approach [23]. In view of this, the historical training data is as well chosen as historical data can be. Furthermore, the NEMS evaluates how severe or nursing intensive a patient is, independently of the diagnosis. Therefore, using is as a covariate in prediction is likely to mitigate confounders between training data and COVID-19 dataset. Second, a limitation is imposed by the use of MDSi as training dataset because the analysis is then constrained to the relatively few variables contained in MDSi. Clearly, there are further relevant predictors for COVID-19 patients. However, most of them concern mortality and not LoS, for example, coagulation status. These values are available in the RISC-19-ICU registry but not in the MDSi training data. Furthermore, we believe that a successful model for probabilistic predictions of LoS should rely on values that are routinely recorded and available early after hospitalization such as SAPS II and NEMS. Since they are compound variables, they are informative for the LoS. If training data sets with more covariates are available, the DIM model we propose in Section 2.2 could be adapted to variables specific to COVID-19 patients. This may lead to an increase in predictive skill. Third, there is possibly a bias towards a longer predicted LoS because of the data sampling process. We have assessed whether the patients with missing LoS value in the RISC-19-ICU registry have a substantially different distribution of covariate values than the patients with valid LoS value. This is not the case which is an indication that many of them, rather than having a censored LoS, have indeed not been updated. We have also repeated all of our analyses on the COVID-19 dataset restricted to patients with admission date before April 5, 2020. Here, the

update and the censoring problem should be less. Qualitatively, we obtained the same results as the ones reported here. Nevertheless, it should be kept in mind that some of the very long LoS are likely to be censored in either case. Fourth, LoS is often not only dependent on epidemiological and physiologic variables but additionally on ICU resources, therapeutic restriction policies [24] and withdrawal strategies (https://www.samw.ch/de/Ethik/Themen-A-bis-Z/Intensivmedizin.html). Our forecasts predict a longer LoS compared with the observed LOS overall and in almost any patient subgroups after 25 days. This may be due to an earlier withdrawal of the intensive therapy compared to ARDS, especially in shortage of ICU resources. However we did not find any significant difference in LoS distribution between two regions of Switzerland with diverse ICU strain.

## 5 Conclusion

A new semiparametric model permits calibrated and informative probabilistic prediction of LoS of individual patients with severe COVID-19 in ICUs, given covariate information. These predictions would allow to simulate stochastic models for bed occupation in ICUs under different scenarios for the case mix. These scenarios could be different projections for the rate at which COVID-19 patients and other patients arrive in the ICUs.

## Supporting information

**S1 Appendix. Additional information about probabilistic forecasting, censoring of LoS in the COVID-19 dataset, and supplementary figures.**
(PDF)

**S1 Data. Minimal dataset to replicate the results of this study.**
(ZIP)

**S1 Code. Sample data and code to illustrate the computation and usage of probabilistic length of stay forecasts.**
(ZIP)

## Acknowledgments

The authors are grateful to the Swiss Society of Intensive Care Medicine for providing access to the MDSi.

RISC-19-ICU Investigators of Switzerland: RISC-19-ICU registry, University of Zurich (Hilty Matthias P, MD; Wendel Garcia Pedro D, MSc; Schüpbach Reto A, MD; Thierry Fumeaux, MD; Jonathan Montomoli, MD, PhD; Philippe Guerci, MD); Klinik für Operative Intensivmedizin, Kantonsspital Aarau, Aarau (Rolf Ensner, MD); Intensivstation, Kantonsspital Schaffhausen, Schaffhausen (Nadine Gehring, MD); Institut fuer Anesthaesie und Intensivmedizin, Zuger Kantonsspital AG, Baar (Peter Schott, MD; Severin Urech, MD); Department Intensivmedizin, Universitaetsspital Basel, Basel (Martin Siegemund, MD; Nuria Zellweger); Intensivmedizin, St. Claraspital, Basel (Adriana Lambert, MD; Lukas Merki, MD); Department of Intensive Care Medicine, University Hospital Bern, Inselspital, Bern (Marie-Madlen Jeitziner, RN, PhD; Beatrice Jenni-Moser, RN, MSc); Department Intensive Care Medicine, Spitalzentrum Biel, Biel (Marcus Laube, MD); Interdisziplinäre Intensivstation, Spital Bülach, Bülach (Bernd Yuen, MD; Thomas Hillermann, MD); Intensivstation, Regionalspital Emmental AG, Burgdorf (Petra Salomon, MD; Iris Drvaric, MD); Intensivmedizin, Kantonsspital Graubünden, Chur (Frank Hillgaertner, MD; Marianne Sieber); Institut fuer Anaesthesie und Intensivmedizin, Spital Thurgau, Frauenfeld (Alexander Dullenkopf, MD; Lina Petersen, MD);

Soins Intensifs, Hopital cantonal de Fribourg, Fribourg (Hatem Ksouri, MD, PhD; Govind Oliver Sridharan, MD); Division of Intensive Care, University Hospitals of Geneva, Geneva (Sara Cereghetti, MD; Filippo Boroli, MD; Jerome Pugin, MD, PhD); Division of Neonatal and Pediatric Intensive Care, University Hospitals of Geneva, Geneva (Serge Grazioli, MD; Peter C. Rimensberger, MD); Intensivstation, Spital Grabs, Grabs (Christian Bürkle, MD); Institut für Anaesthesiologie Intensivmedizin & Rettungsmedizin, See-Spital Horgen & Kilchberg, Horgen (Julien Marrel, MD; Mirko Brenni, MD); Soins Intensifs, Hirslanden Clinique Cecil, Lausanne (Isabelle Fleisch, MD; Jerome Lavanchy, MD); Anaesthesie und Intensivmedizin, Kantonsspital Baselland, Liestal (Anja Baltussen Weber, MD; Peter Gerecke, MD; Andreas Christ, MD); Dipartimento Area Critica, Clinica Luganese Moncucco, Lugano (Romano Mauri, MD; Samuele Ceruti, MD); Interdisziplinaere Intensivstation, Spital Maennedorf AG, Maennedorf (Katharina Marquardt, MD; Karim Shaikh, MD); Institut fuer Anaesthesie und Intensivmedizin, Spital Thurgau, Münsterlingen (Thomas Neff, MD; Tobias Hübner, MD); Intensivmedizin, Schweizer Paraplegikerzentrum Nottwil, Nottwil (Hermann Redecker, MD); Soins intensifs, Groupement Hospitalier de l'Ouest Lémanique, Hôpital de Nyon, Nyon (Thierry Fumeaux, MD; Mallory Moret-Bochatay, MD); Intensivmedizin & Intermediate Care, Kantonsspital Olten, Olten (Michael Studhalter, MD); Intensivmedizin, Spital Oberengadin, Samedan (Michael Stephan, MD; Jan Brem, MD); Anaesthesie Intensivmedizin Schmerzmedizin, Spital Schwyz, Schwyz (Daniela Selz, MD; Didier Naon, MD); Medizinische Intensivstation, Kantonsspital St. Gallen, St. Gallen (Gian-Reto Kleger, MD); Departement of Anesthesiology and Intensive Care Medicine, Kantonsspital St. Gallen, St. Gallen (Miodrag Filipovic, MD; Urs Pietsch, MD); Paediatric Intensive Care Unit, Children's Hospital of Eastern Switzerland, St. Gallen (Bjarte Rogdo, MD; Andre Birkenmaier, MD); Departement for intensive care medicine, Kantonsspital Nidwalden, Stans (Anette Ristic, MD; Michael Sepulcri, MD); Intensivstation, Spital Simmental-Thun-Saanenland AG, Thun (Antje Heise, MD); Klinik für Anaesthesie und Intensivmedizin, Spitalzentrum Oberwallis, Visp (Friederike Meyer zu Bentrup, MD, MBA); Service d'Anesthesiologie, EHNV, Yverdon- les-Bains (Marilene Franchitti Laurent, MD; Jean-Christophe Laurent, MD); Institute of Intensive Care Medicine, University Hospital Zurich, Zurich (Philipp Bühler, MD; Silvio Brugger, MD, PhD; Jan Bartussek, PhD; Martina Maibach, PhD; Annelies Zinkernagel, MD, PhD, Dorothea Heuberger, PhD; Srikanth Mairpady Shambat, PhD); Interdisziplinaere Intensivstation, Stadtspital Triemli, Zurich (Patricia Fodor, MD; Pascal Locher, MD; Giovanni Camen, MD); Abteilung für Anaesthesiologie und Intensivmedizin, Hirslanden Klinik Im Park, Zürich (Tomislav Gaspert, MD; Marija Jovic, MD); Institut für Anaesthesiologie und Intensivmedizin, Klinik Hirslanden, Zurich (Christoph Haberthuer, MD; Roger F. Lussman, MD).

## Author Contributions

**Conceptualization:** Alexander Henzi, Gian-Reto Kleger, Johanna F. Ziegel.

**Data curation:** Matthias P. Hilty, Pedro D. Wendel Garcia.

**Formal analysis:** Alexander Henzi.

**Investigation:** Alexander Henzi, Gian-Reto Kleger, Johanna F. Ziegel.

**Methodology:** Alexander Henzi, Gian-Reto Kleger, Johanna F. Ziegel.

**Project administration:** Gian-Reto Kleger, Johanna F. Ziegel.

**Software:** Alexander Henzi, Matthias P. Hilty, Pedro D. Wendel Garcia.

**Supervision:** Johanna F. Ziegel.

**Validation:** Gian-Reto Kleger, Matthias P. Hilty, Pedro D. Wendel Garcia.

**Visualization:** Alexander Henzi.

**Writing – original draft:** Johanna F. Ziegel.

**Writing – review & editing:** Alexander Henzi, Gian-Reto Kleger, Matthias P. Hilty, Pedro D. Wendel Garcia, Johanna F. Ziegel.

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
