## [Decision Letter · Decision Letter 0]

11 Dec 2020

PONE-D-20-30127

Probabilistic analysis of COVID-19 patients’ individual length of stay in Swiss intensive care units

PLOS ONE

Dear Dr. Ziegel,

Thank you for submitting your manuscript to PLOS ONE. After careful consideration, we feel that it has merit but does not fully meet PLOS ONE’s publication criteria as it currently stands. Therefore, we invite you to submit a revised version of the manuscript that addresses the points raised during the review process.

The paper is based on an interesting topic, the statistical analysis has been performed appropriately, but there many concerns authors should correct and some concepts clarified to make the manuscript suitable for publication. This paper could be very difficult to be interpreted by a physician without mathematical experience. Authors should simplify the interpretations of results. 

There are no conflicts beteween reviews.Please answer to all the questions moved by reviewers. 

We look forward to receiving your revised manuscript.

Kind regards,

Martina Crivellari

Academic Editor

PLOS ONE

Journal Requirements:

2. In your ethics statement in the manuscript and in the online submission form, please provide additional information about the patient records/samples used in your retrospective study.

Specifically, please ensure that you have discussed whether all data/samples were fully anonymized before you accessed them.

If patients provided informed written consent to have data/samples from their medical records used in research, please include this information.

Reviewers' comments:

Reviewer's Responses to Questions

**Comments to the Author**

1. Is the manuscript technically sound, and do the data support the conclusions?

Reviewer #1: Partly

Reviewer #2: Partly

Reviewer #3: Partly

2. Has the statistical analysis been performed appropriately and rigorously? 

Reviewer #1: I Don't Know

Reviewer #2: I Don't Know

Reviewer #3: Yes

3. Have the authors made all data underlying the findings in their manuscript fully available?

Reviewer #1: No

Reviewer #2: No

Reviewer #3: Yes

4. Is the manuscript presented in an intelligible fashion and written in standard English?

Reviewer #1: Yes

Reviewer #2: Yes

Reviewer #3: Yes

5. Review Comments to the Author

Reviewer #1: This paper attempts to provide a way to estimate LoS for individual patients, based on characteristics available within the first 24 hours of admission. I think that this is a very valuable contribution, as it can help with health care planning and provide more accurate estimates for when hospital capacity may be exceeded.

Although I think this is a nice paper in principal, I don't think enough information has been provided for me to adequately review the methods and results. As it stands I don't think the methods are clear, and the interpretation of the results is difficult to follow. I have tried to highlight below areas that I think could be made clearer:

Line 13, you talk about probabilistic predictions, but even with your definition it is not clear what this means. How is the uncertainty of the LoS quantified? This ties in with Figure 1, which I also don't think is clear. There is very little information provided in the figure legend or text about figure 1, and it is difficult to interpret. As I understand, each vertical line indicates how long someone actually stays in hospital, so patient 4 stayed in for 40 days? And then patient 3 was released from ICU on day ±1, but had a very low possibility of doing so (±0.05)? I think more of an explanation is the figure legend and text is required to adequately explain this figure, maybe even just proving a small example as I have done above would help. I think it's a nice figure which if adequately explained in the legend and text would help to clarify the aim/ methods of the paper. At the moment I don't think your overall aim is clear.

It is also not made clear in the introduction how having the CDF would be used. Who are you expecting to use your results, and how? It says in the abstract that individuals with long LoS could be discovered early, indicating that maybe this is for hospital planning purposes? Also, going back to figure 1, patient 4 has the longest LoS, but it is not clear precisely what indicates that patient 4 is going to have the longest LoS from the Forecast CDF. So difficult to see how the CDF is going to indicate long los.

Methods:

I understand that you are not able to release your data, however, there is no reason that the code could not be released, along with a brief description of the datasets you have available (could even consider creating some dummy datasets). This would allow others to understand your methods more clearly, and be able to repeat your analysis.

From your methods, it's not adequately explained what the difference is between ECDF and CDF. In addition, in the results you discuss ECDF, and show plots comparing them to CDF, but I think providing a brief summary at the beginning of the results would aid interpretation.

Results:

You talk about how this is done on an individual level, and yet in Figures 2-4 your provide figures for one overall CDF. So is this a CDF for the whole dataset? How have you combined them?

Line 138 you talk about panel c of figure 2, but there is no panel c.

Overall, I think this is an interesting idea and concept, but in it's current format I don't think their methods are reproducible and their results are not easy to follow. I think they need to be clear who is there target audience, is it for mathematical modellers or people with a more clinical background? Clinicians or anyone with a non-technical background would struggle to know what to do with this information. However, I do think it is highly relevant, so I hope that the authors are able to revise their manuscript to make things clearer. I think being able to predict who is going to spend a long time on ICU is of great value. Best of luck with the submission.

Reviewer #2: In this paper the authors developed a new semi parametric distributional index model that should provide a probabilistic prediction of ICU length of stay 24 hours after admission for COVID-19 patients.

The model is based on 4 covariates: age, gender, SAPS II and NEMS.

According to the authors these parameters were the only possible choice.

I wonder if and how covariates dependance affects the model. Particularly:

- age is included in SAPS II

- both SAPS II and NEMS are expression of the severity of patients status, is this model performing better than a simpler model including just SAPS II?

On the opposite, I wonder if including more specific variables, such as the coagulation status would provide better predictions.

For the quality and soundness of the statistical analysis, I do not have the skills to judge, and an expert’s opinion is needed.

Minor comments:

* Minor language revision is needed, as the manuscript contains several typos

Reviewer #3: The authors proposed a new semi-parametric model to predict individual ICU LOS even though prediction of LOS at the patient level is difficult and none of the available models were reliable. The model was fully demonstrated in two of method papers showing that it provides more information than classical models.

In Figure 1, predicted CDFs of LOS and the corresponding true LOS were depicted. In reality, patient 3 had the shortest LOS and patients 1 and 2 had longer LOS. However, the forecasted CDF for those three patients were very close to each other. It looks like the proposed model doesn’t have the ability to provide satisfactory forecasts at the patient level.

The authors mentioned LOS of a patient in ICU also depends on the characteristics and policies of the ICU. The COVID-19 pandemic placed a significant burden on healthcare systems. It induced unprecedented strain on ICU resources. It brought systematic error of prediction by using patients who were diagnosed of ARDS in 6 concurrent years as the training cohort. Because both measured and unmeasured confounders wound be unbalanced between training and COVID-19 cohorts.

It may be helpful to present comparisons of other measured confounders and patients’ dispositions between cohorts.

Patients who were admitted to ICU had two dispositions, discharged or dead. It looks like the authors included both discharged and dead patients in analysis. Was patients’ disposition considered differently in model? If not, can the authors clarify?

In line 153, the authors reported patients who were greater than 80 years of age have much shorter LOS. Is it because in-ICU mortality rate was higher for this age-subgroup?

The authors used cubic spline for continuous variables age, SAPS II, and NEMS in regression. Can the authors justify their decision to use cubic spline and how the assumptions of such a model were considered? Did the authors conducted tests for curvature and tests for significance of each curve?

Figure 1: The color and pattern of lines confused me. Why did you use green lines for patients 1, 2, and 3 and red lines for patient 4?

The paragraph of censoring of LOS in Appendix should be reported as a limitation in the discussion section.

6. PLOS authors have the option to publish the peer review history of their article (what does this mean?). If published, this will include your full peer review and any attached files.

Reviewer #1: No

Reviewer #2: No

Reviewer #3: No

---

## [Author Response · Author response to Decision Letter 0]

12 Jan 2021

Detailed responses to all comments of the editor and the reviewers are provided in the pdf-file "response_Covid19_2020.12.17.pdf".

---

## [Decision Letter · Decision Letter 1]

29 Jan 2021

PONE-D-20-30127R1

Probabilistic analysis of COVID-19 patients’ individual length of stay in Swiss intensive care units

PLOS ONE

Dear Dr. Ziegel,

Thank you for submitting your manuscript to PLOS ONE. After careful consideration, we feel that it has merit but does not fully meet PLOS ONE’s publication criteria as it currently stands. Therefore, we invite you to submit a revised version of the manuscript that addresses the points raised during the review process.

 ACADEMIC EDITOR:  Most of comments have been addressed. I've asked a check to the statistician, see conclusions below. Please modify the manuscript as suggested to make it definitely suitable for publication. 

 Please submit your revised manuscript by  Feb 15th. If you will need more time than this to complete your revisions, please reply to this message or contact the journal office at plosone@plos.org. Please include the following items when submitting your revised manuscript:

We look forward to receiving your revised manuscript.

Kind regards,

Martina Crivellari

Academic Editor

PLOS ONE

Reviewers' comments:

Reviewer's Responses to Questions

**Comments to the Author**

1. If the authors have adequately addressed your comments raised in a previous round of review and you feel that this manuscript is now acceptable for publication, you may indicate that here to bypass the “Comments to the Author” section, enter your conflict of interest statement in the “Confidential to Editor” section, and submit your "Accept" recommendation.

Reviewer #3: (No Response)

2. Is the manuscript technically sound, and do the data support the conclusions?

Reviewer #3: Yes

3. Has the statistical analysis been performed appropriately and rigorously? 

Reviewer #3: Yes

4. Have the authors made all data underlying the findings in their manuscript fully available?

Reviewer #3: Yes

5. Is the manuscript presented in an intelligible fashion and written in standard English?

Reviewer #3: Yes

6. Review Comments to the Author

Reviewer #3: The authors have addressed most of my comments and concerns.

The reviewer still worry about prediction accuracy. In figure 1, patient 3 had the shortest LoS but the forecast CDF curve was in the middle. The order of realized LoS from the shortest to the longest were (patient 3 < patient 2 < patient 1 < patient 4). But the predicted probabilities were in different order.

In the legend of figure 1, you said if a patient left on day t, the predictive CDF would jump from 0 to 1 at t. However, the CDFs in figure 1 didn’t do so. Patient left on day 1, but the corresponding CDF didn’t jump to 1.

The reviewer suggested to draw a 2-D scatter plot to indicate prediction accuracy. The x-axis is the observed values of the LoS and the y-axis is the probability that the respective patient would discharged on the realized day. Each dot represents each patient. For example, patient 1 was discharged/dead on day 20 and the probability that he/she would discharged on day 20 was about 80%. So the dot of patient 1 should be located at (x = 20, y = 80%).

7. PLOS authors have the option to publish the peer review history of their article (what does this mean?). If published, this will include your full peer review and any attached files.

Reviewer #3: No

---

## [Author Response · Author response to Decision Letter 1]

1 Feb 2021

Please see the attached file "Response to Reviewers".

---

## [Editor Report · Decision Letter 2]

4 Feb 2021

Probabilistic analysis of COVID-19 patients’ individual length of stay in Swiss intensive care units

PONE-D-20-30127R2

Dear Dr. Ziegel,

We’re pleased to inform you that your manuscript has been judged scientifically suitable for publication and will be formally accepted for publication once it meets all outstanding technical requirements.

Kind regards,

Martina Crivellari

Academic Editor

PLOS ONE
---

## [Editor Report · Acceptance letter]

10 Feb 2021

PONE-D-20-30127R2 

Probabilistic analysis of COVID-19 patients' individual length of stay in Swiss intensive care units 

Dear Dr. Ziegel:

I'm pleased to inform you that your manuscript has been deemed suitable for publication in PLOS ONE. Congratulations! Your manuscript is now with our production department. 

Kind regards, 

on behalf of

Dr. Martina Crivellari 

Academic Editor

PLOS ONE